# Cost-utility analysis of de-escalating biological disease-modifying anti-rheumatic drugs in patients with rheumatoid arthritis

**Benjamin Birkner**[1]*, **Jürgen Rech**[2], **Tom Stargardt**[1]

**1** Hamburg Center for Health Economics (HCHE), Universität Hamburg, Hamburg, Germany, **2** Friedrich-Alexander-University Erlangen-Nürnberg (FAU), Department of Internal Medicine 3 – Rheumatology and Immunology, Universitätsklinikum Erlangen, Erlangen, Germany

* benjamin.birkner@uni-hamburg.de

**Data Availability Statement:** All relevant data are within the paper and its Supporting Information files.

## Abstract

### Objective

Recent guideline updates have suggested de-escalating DMARDs when patients with rheumatoid arthritis achieve remission or low disease activity. We aim to evaluate whether it is cost-effective to de-escalate the biological form of DMARDs (bDMARDs).

### Methods

Using a Markov model, we performed a cost-utility analysis for RA patients on bDMARD treatment. We compared continuing treatment (standard care) to a tapering approach (i.e., an immediate 50% dose reduction), withdrawal (i.e., an immediate 100% dose reduction) and tapering followed by withdrawal of bDMARDs. The parametrization is based on a comprehensive literature review. Results were computed for 30 years with a cycle length of three months. We applied the payer's perspective for Germany and conducted deterministic and probabilistic sensitivity analyses.

### Results

Tapering or withdrawing bDMARD treatment resulted in ICERs of €526,254 (incr. costs -78,845, incr. QALYs -0.1498) or €216,879 (incr. costs -€121,691, incr. QALYs -0.5611) compared to standard care. Tapering followed by withdrawal resulted in a loss of 0.4354 QALYs and savings of €107,969 per patient, with an ICER of €247,987. Deterministic sensitivity analysis revealed that our results remained largely unaffected by parameter changes. Probabilistic sensitivity analysis suggests that tapering, withdrawal and tapering followed by withdrawal were dominant in 39.8%, 28.2% and 29.0% of 10,000 iterations.

### Conclusion

Our findings suggest that de-escalating bDMARDs in patients with RA may result in high cost savings but also a decrease in quality of life compared to standard care. If decision

**Funding:** The authors received no specific funding for this work.

**Competing interests:** The authors have declared that no competing interests exist.

makers choose to implement de-escalation in daily practice, our results suggest the tapering approach.

# 1 Introduction

Rheumatoid Arthritis (RA) is one of the most prevalent chronic inflammatory diseases in North America and Europe [1]. In 2013, the prevalence of RA in Germany was an estimated 1.4%[2]. Treatment follows a treat-to-target approach, which aims for persistent clinical remission or at least low disease activity[3]. Pharmacological treatment consists of analgesics, non-steroidal anti-inflammatory drugs (NSAIDs), glucocorticoids, conventional synthetic disease-modifying antirheumatic drugs (csDMARDs), targeted synthetic DMARDs (tsDMARDs), biological DMARDs (bDMARDs), or combination of these.

The number of patients treated with the highly effective bDMARDs has increased continuously in the US, Europe and Australia since their introduction in the early 2000s[4]. The share of RA patients treated with bDMARDs in Germany, for example, increased from 4.4% in 2002 [5] to 27.3% in 2014[6]. Annual mean drug costs per RA patient more than doubled in Germany during this period[5]. In 2016, the 41.3 million defined daily doses of bDMARDs prescribed in Germany[7] accounted for more than two billion euros in health care expenditure.

Given the high cost of long-term bDMARD use, international treatment guidelines have recommended tapering bDMARDs and csDMARDs for patients with RA in stable remission or who have low disease activity[3,8–10]. Although there is evidence from recent systematic literature reviews of promising clinical outcomes in patients who have de-escalated bDMARDs [11–13], health-economic evaluations of de-escalation approaches are scarce. While previous research has shown that de-escalation approaches lead to substantial savings on the cost side, the impact on quality of life (QoL) is much less clear. On the one hand, Vanier et al.[14] and Tran-Duy et al.[15] identified a trade-off between costs and QoL from the French and from the Dutch perspective, respectively. On the other hand, Kobelt[16] found de-escalation to be dominant, i.e., de-escalation to increase QoL. However, all three studies used data from a single trial only (Vanier et al.: STRASS trial, Tran-Duy et al.: POET trial, Kobelt et al.: PRESERVE trial).

In the present study, we focused on bDMARDs due to their high economic impact and modelled three approaches to their de-escalation: tapering (defined as an immediate 50% dose reduction), withdrawal (defined as an immediate 100% dose reduction) and tapering for at least six months followed by withdrawal.

In contrast to previous literature[14–16], our analysis (a) considers three different approaches to de-escalating bDMARDs, (b) pools all available evidence from the literature and (c) applies a long-term perspective to capture variations in disease activity. Furthermore, we also generate evidence from the payer's perspective for Germany. To do so, we developed a Markov model with a time-horizon of 30 years and used random-effects pooling of clinical data for model parameter estimation.

# 2 Materials and methods

To model the de-escalation of bDMARDs from the payer's perspective, we compared the costs and outcomes of standard care to those of three possible interventions: (a) tapering, (b) withdrawal and (c) tapering followed by withdrawal. We defined standard care as combination

therapy comprising one bDMARD, one csDMARD and concomitant treatment with NSAIDs and glucocorticoids.

We used a cohort-based Markov model because this allowed us to capture recurring events in disease activity over a long time horizon. We included costs and quality-adjusted life years (QALYs) as outcomes and discounted these at 3%. We chose a cycle length of three months and a time horizon of 30 years (120 cycles), applying half-cycle correction. Our hypothetical starting cohort consists of 1,000 patients in the age of 45 years. Initial disease activity of RA is medium to high as guidelines suggest adding bDMARDs at this stage[3,8]. For calculations we used the heemod package for R[17] and Microsoft Excel.

## 2.1 Model structure

We defined the Markov states in our model according to disease activity: clinical remission (REM), low disease activity (LDA), medium to high disease activity (M/HDA) and death. After each cycle, patients remained in their current state, entered a state with increased or decreased disease activity, or died (Fig 1).

Patients with persistent REM or LDA after 3 months (one cycle) continued to states for sustained remission (S_REM) or low disease activity (S_LDA). Patients with sustained or

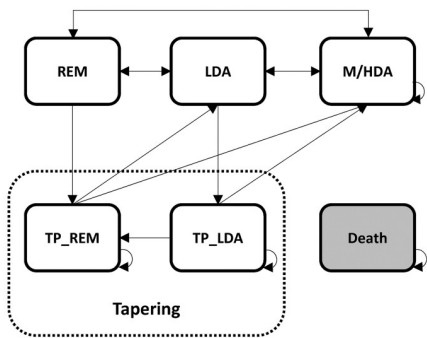

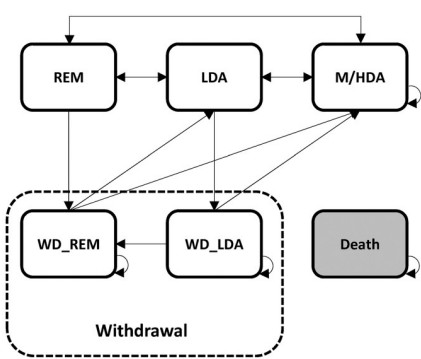

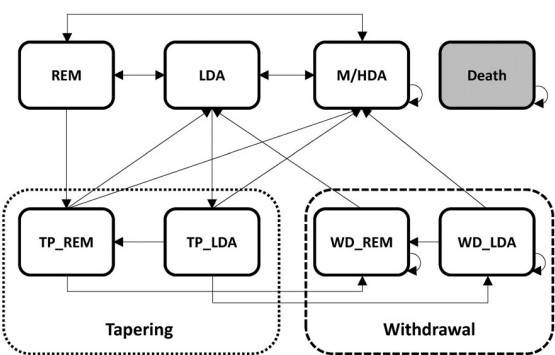

**Fig 1. Model structure.** (A) standard care without dose adjustments, (B) tapering, (C) withdrawal, (D) tapering followed by withdrawal of bDMARDs. REM: remission, LDA: low disease activity, M/HDA: medium to high disease activity, TP_REM: remission under tapering, TP_LDA: low disease activity under tapering, WD_REM: remission under immediate withdrawal, WD_LDA: low disease activity under immediate withdrawal. Arrows indicate possible transitions between states at each model cycle of 6 months.

improved disease activity after six months (two cycles) entered the de-escalation states. Depending on the de-escalation strategy, patients either received a 50% reduced dose of bDMARDs (states: TP_REM, TP_LDA) or were completely withdrawn from bDMARDs immediately (states: WD_REM, WD_LDA). For the approach that involved dose reduction followed by withdrawal, patients had to remain stable in the tapering state for at least six months before entering the withdrawal state (states: S_TP_REM, S_TP_LDA). Model states depicting sustained disease activity rely on the same transition probabilities as their origin states. In cases where a patient's disease activity worsened under de-escalation, patients were transferred back to the LDA or M/HDA state and resumed bDMARDs treatment at the full dose. These patients were eligible for another round of de-escalation in the event that they subsequently achieved remission or low-disease activity for six months again. There was no restriction on the number of times patients could be transferred back to the LDA or M/HDA states and subsequently become eligible for another round of de-escalation. Lastly, in all of the states, all individuals were at risk of all-cause mortality.

## 2.2 Parametrization

**2.2.1 Transition probabilities.** The transition probabilities in our model were estimated by applying random effects pooling [18] on 14 clinical studies[19–32] identified in four systematic reviews[11–13,33], as well as eight additional studies[34–40] found in a literature search conducted in December 2017 using PubMed and Google Scholar to update the systematic reviews. To be included in our analysis, a study had to be a clinical trial or an observational study, measure disease activity using DAS28 based on erythrocyte sedimentation rate (ESR) or C-reactive protein (CRP), definition of remission as DAS28<2.6 or report results by severity (i.e. DAS8<2.6, DAS28<3.2) and involve pharmacological treatment comprising a combination of one csDMARD and one bDMARD administered in at least one of our settings. A detailed overview of our study pool can be found S2 Table.

We took a three step approach to estimate transition probabilities. First, we categorized studies according to the baseline DAS28 of the underlying study populations (REM: DAS28 < 2.6, LDA: DAS28 < 3.2, M/HDA: DAS28 > 3.2).

Second, we extracted the number of patients per category after 3 months of follow-up. Where these were not available we used the number of patients of the closest follow-up period. Third, we transformed the number of patients per state to transition probabilities from state to state. We calculated transition probabilities per cycle by transforming the transition probability for each study to a constant rate and then further to a transition probability based on our cycle length [41]. We used the following formula to apply both transformations in one step $(p_r = 1 - (1 - p_s)^{\frac{t}{s}})$, where $t$ denotes our cycle length of three months and $s$ the study's duration in months.

We applied random effects pooling as a final step to estimate transition probabilities for each of the model states (Table 1, pooled estimate). By doing so we considered the studies' outcomes with respect to individual sample size, reducing the possible impact of biased study results. An overview of the used study populations, calculated transition probabilities as well as pooled estimates of transition probabilities are shown in Table 1.

**2.2.2 Mortality.** Data on mortality were based on WHO 2015 all-cause mortality for Germany. We allowed these to vary over time to reflect the ageing of the cohort. A weighted average of gender-specific all-cause mortality ensured a 3:1 ratio of female to male patients to reflect the gender distribution of RA patients in Germany[6]. We assumed a 2.34 times higher mortality risk for patients with moderate/severe RA compared to the general population based on findings for a German cohort[42].

**Table 1. Transition probabilities.**

| Study | Year | Journal | Substance | Group | Design | Nᵃ | REM | LDA | M/HDA | follow-up | Transformed six-month probabilities | | |
|---|---|---|---|---|---|---|---|---|---|---|---|---|---|
| **Standard Care** | | | | | | | | | | | | | |
| **Starting state: REM** | | | | | | | | | | | REM → REM | REM → LDA | REM → M/HDA |
| Emery et al. | 2014 | New Engl J Med | ETA | CG | RCT | 63 | 55 | 3 | 5 | 10 | 0.8413 | 0.0794 | 0.0794 |
| Tanaka et al. | 2015 | Ann Rheum Dis | ADA | CG | OC | 23 | 19 | 2 | 2 | 12 | 0.9550 | 0.0225 | 0.0225 |
| Haschka et al. | 2016 | Ann Rheum Dis | TNF | CG | RCT | 38 | 33 | 5 | 0 | 6 | 0.8684 | 0.1316 | 0.0000 |
| **Pooled estimate** | . | | | . | . | . | . | . | . | . | **0.8911** | **0.0666** | **0.0491** |
| **Starting state: LDA** | | | | | | | | | | | LDA → REM | LDA → LDA | LDA → M/HDA |
| Smolen et al. | 2013 | Lancet | ETA | CG | RCT | 181 | 134 | 32 | 15 | 12 | 0.2862 | 0.6925 | 0.0214 |
| Smolen et al. | 2014 | Lancet | ADA | CG | RCT | 96 | 90 | 6 | 0 | 12 | 0.5000 | 0.5000 | 0.0000 |
| van Herwaarden et al. | 2015 | BMJ Brit Med J | ADA, ETA | CG | RCT | 59 | 48 | 6 | 5 | 9 | 0.4287 | 0.5422 | 0.0291 |
| Keystone et al. | 2016 | Rheumatology | ETA | IG | RCT | 47 | 24 | 6 | 17 | 18 | 0.1123 | 0.8156 | 0.0721 |
| **Pooled estimate** | . | | | . | . | . | . | . | . | . | **0.3291** | **0.6390** | **0.0256** |
| **Starting state: M/HDA** | | | | | | | | | | | M/HDA → REM | M/HDA → LDA | M/HDA → M/HDA |
| Kremer | 2005 | Arthritis Rheumatol | ABA | IG | RCT | 99 | 17 | 12 | 70 | 3 | 0.1717 | 0.1212 | 0.7071 |
| Westhovens et al. | 2006 | Arthritis Rheumatol | INF | IG | RCT | 663 | 216 | 371 | 76 | 6 | 0.1789 | 0.3364 | 0.4847 |
| Emery et al. | 2008 | Lancet | ETA | IG | RCT | 221 | 132 | 38 | 51 | 12 | 0.2034 | 0.0461 | 0.7505 |
| Schiff et al. | 2008 | Ann Rheum Dis | ABA | IG | RCT | 147 | 14 | 17 | 116 | 6 | 0.0488 | 0.0596 | 0.8916 |
| Schiff et al. | 2008 | Ann Rheum Dis | INF | IG | RCT | 152 | 20 | 20 | 112 | 6 | 0.0681 | 0.0681 | 0.8638 |
| Ruubert-Roth | 2010 | Rheumatology | RITU | IG | RCT | 314 | 44 | 31 | 239 | 12 | 0.0370 | 0.0257 | 0.9373 |
| Kim et al. | 2012 | Int J Rheum Dis | ETA | IG | RCT | 193 | 82 | 30 | 81 | 4 | 0.3396 | 0.1190 | 0.5414 |
| Kavanaugh et al. | 2013 | Ann Rheum Dis | ADA | IG | RCT | 466 | 175 | 67 | 224 | 6 | 0.2098 | 0.0747 | 0.7156 |
| Smolen et al. | 2013 | Lancet | ETA | CG | RCT | 757 | 525 | 152 | 80 | 9 | 0.3258 | 0.0720 | 0.6022 |
| Yoo et al. | 2013 | Ann Rheum Dis | INF | IG | RCT | 246 | 46 | 41 | 159 | 4 | 0.1438 | 0.1278 | 0.7284 |
| Yoo et al. | 2013 | Ann Rheum Dis | INF | CG | RCT | 249 | 44 | 29 | 176 | 4 | 0.1357 | 0.0887 | 0.7756 |
| Dougados | 2014 | Ann Rheum Dis | TOC | IG | RCT | 277 | 112 | 59 | 106 | 6 | 0.2282 | 0.1129 | 0.6589 |
| Horslev-Petersen et al. | 2014 | Ann Rheum Dis | ADA | IG | RCT | 81 | 66 | 5 | 10 | 12 | 0.3440 | 0.0158 | 0.6402 |
| Machado et al. | 2014 | JCR-J Clin Rheumatol | ETA | IG | RCT | 269 | 70 | 61 | 138 | 6 | 0.1399 | 0.1207 | 0.7394 |
| Nam et al. | 2014 | Ann Rheum Dis | INF | CG | RCT | 55 | 22 | 7 | 26 | 4 | 0.3338 | 0.0971 | 0.5847 |
| Schiff et al. | 2014 | Ann Rheum Dis | ABA, ADA | IG | RCT | 274 | 138 | 51 | 85 | 12 | 0.1606 | 0.0502 | 0.7892 |
| Schiff et al. | 2014 | Ann Rheum Dis | ABA, ADA | CG | RCT | 269 | 137 | 64 | 68 | 12 | 0.1630 | 0.0657 | 0.7713 |
| Keystone et al. | 2016 | Rheumatology | ETA | IG | RCT | 58 | 8 | 9 | 41 | 18 | 0.0244 | 0.0277 | 0.9479 |
| **Pooled estimate** | . | | | . | . | . | . | . | . | . | **0.1760** | **0.0895** | **0.7305** |
| **Tapering** | | | | | | | | | | | | | |
| **Starting state: REM** | | | | | | | | | | | TP_REM → TP_REM | TP_REM → LDA | TP_REM → M/HDA |
| Haschka et al. | 2016 | Ann Rheum Dis | TNF | IG | RCT | 36 | 29 | 4 | 3 | 3 | 0.8056 | 0.1111 | 0.0833 |
| Haschka et al. | 2016 | Ann Rheum Dis | TNF | IG | RCT | 27 | 25 | 0 | 2 | 3 | 0.9259 | 0.0000 | 0.0741 |
| **Pooled estimate** | . | | | . | . | . | . | . | . | . | **0.8733** | **0.1111** | **0.0791** |
| **Starting state: LDA** | | | | | | | | | | | TP_LDA → REM | TP_LDA → TP_LDA | TP_LDA → M/HDA |
| Smolen et al. | 2013 | Lancet | ETA | IG | RCT | 175 | 121 | 38 | 16 | 12 | 0.2547 | 0.7216 | 0.0237 |
| **Pooled estimate** | . | | | . | . | . | . | . | . | . | **0.2547** | **0.7216** | **0.0237** |
| **Withdrawal** | | | | | | | | | | | | | |
| **Starting state: REM** | | | | | | | | | | | WD_REM → WD_REM | WD_REM → LDA | WD_REM → M/HDA |

*(Continued)*

**Table 1.** (Continued)

| Study | Year | Journal | Substance | Group | Design | Nᵃ | REM | LDA | M/HDA | follow-up | Transformed six-month probabilities | | |
|---|---|---|---|---|---|---|---|---|---|---|---|---|---|
| Emery et al. | 2014 | New Engl J Med | ETA | IG | RCT | 65 | 43 | 8 | 14 | 3 | 0.6615 | 0.1231 | 0.2154 |
| Nam et al. | 2014 | Ann Rheum Dis | ETA | IG | RCT | 40 | 17 | 7 | 16 | 6 | 0.6829 | 0.0917 | 0.2254 |
| Tanaka et al. | 2015 | Ann Rheum Dis | ADA | IG | OC | 52 | 25 | 7 | 20 | 12 | 0.8502 | 0.0355 | 0.1143 |
| Haschka et al. | 2016 | Ann Rheum Dis | TNF | IG | RCT | 27 | 20 | 4 | 3 | 3 | 0.7407 | 0.1481 | 0.1111 |
| **Pooled estimate** | . | | | . | . | . | . | . | . | . | **0.7394** | **0.0843** | **0.1616** |
| **Starting state: LDA** | | | | | | | | | | | WD_LDA → WD_REM | WD_LDA → WD_LDA | WD_LDA → M/HDA |
| Tanaka et al. | 2010 | Ann Rheum Dis | INF | IG | SA | 102 | 44 | 12 | 46 | 12 | 0.1316 | 0.7292 | 0.1392 |
| Smolen et al. | 2013 | Lancet | ETA | IG | RCT | 141 | 58 | 26 | 57 | 12 | 0.1241 | 0.7545 | 0.1215 |
| Smolen et al. | 2014 | Lancet | ADA | IG | RCT | 89 | 67 | 15 | 7 | 12 | 0.2949 | 0.6848 | 0.0203 |
| **Pooled estimate** | . | | | . | . | . | . | . | . | . | **0.1764** | **0.7322** | **0.0905** |

ABA: abatacept, ADA: adalimumab, CG: control group, ETA: etanercept, IG: intervention group, INF: infliximab, LDA: low disease activity, M/HDA: medium or high disease activity, OC: observational cohort, RCT: randomized controlled trial, REM: clinical remission, SA: single-arm trial, TNF: TNF-alpha inhibitors, TOC: tocilizumab, TP: tapering, WD: withdrawal;

ᵃCorrected for drop outs

**2.2.3 Costs.** We considered two categories of direct costs: pharmaceutical and all other direct costs. Pharmaceutical costs consisted of the average costs to the payer of (a) one csDMARD, (b) one bDMARD and (c) concomitant treatment with one NSAID and one glucocorticoid.

First, we subtracted all discounts and co-payments from pharmacy retail prices for 2017. Second, we divided the calculated costs per package by the amount of milligrams per package. We considered prices for 17 bDMARDs, two csDMARDs, one glucocorticoid and two NSAIDs for the calculation. Third, to calculate the required dose in mg per cycle, we multiplied prices per mg by recommended doses from product information by the European Medicine Agency and recommendations from Smolen et al.[43]. We calculated average costs in each category per year as (a) €14,834 for bDMARDs, (b) €1,086 for csDMARDS and €45 for concomitant medication (for detailed information, see S1 Table).

Other direct costs were taken from Huscher et al.[5], who report average direct costs per disease activity state for patients with RA in Germany in 2011. Non-pharmaceutical costs in our study comprised those of hospitalization, rehabilitation, physician visits, joint replacement surgery, physiotherapy and imaging[5]. We inflated costs by 29.6% to reflect 2017 prices by incorporating growth of healthcare expenditure in Germany since 2010. Thus, all other direct costs for patients aged ≤ 64 years amounted to €1,495 (€2,024 for > 65 years) when in REM, €2,316 (€2,421) when in LDA and €3,713 (€3,881) when in M/HDA.

Annual total direct costs per state per patient aged ≤ 64 years were €17,460 (€17,989 for > 65 years) for REM, €18,281 (€18,386) for LDA and €19,678 (€19,846) for M/HDA under standard care. For tapering, we considered only 50% of the bDMARD costs, yielding total direct costs of €10,043 (€10,572) for TP_REM and €10,864 (€10,969) for TP_LDA. For withdrawal, we calculated costs of €2,626 (€3,155) for patients in WD_REM and €3,447 (€3,552) patients in WD_LDA.

**2.2.4 Quality of life.** To measure QALYs, we relied on EQ-5D values by disease state from a Dutch cohort[44]. Depending on disease activity, quality-of-life values were 0.75, 0.71 and 0.60 for patients in any REM, any LDA or the M/HDA state, respectively.

## 2.3 Sensitivity analyses

We addressed parameter uncertainty by conducting deterministic sensitivity analyses and a Monte Carlo simulation with 10,000 iterations. For the one-way deterministic sensitivity analysis, we varied all cost parameters by 45% based on findings by Fautrel et al.[45] for French patients under combination treatment with csDMARDs and bDMARDs. Upper and lower bounds for utilities were based on the reported 95% confidence intervals of Welsing et al.[44], whereas bounds for the transition probabilities were the 95% confidence intervals from the random effects pooling. As upper and lower bounds for the discount rate, we used 6% and 0% (adjusted to 3 month cycle length).

For probabilistic sensitivity analysis, we conducted a Monte Carlo simulation with 10,000 iterations. We chose a gamma distribution for costs[45], Dirichlet distribution for transition probabilities and beta distribution for quality of life[46]. Details of all parameters can be found in Table 2.

To test structural assumptions and further assess heterogeneity in our results, we performed several scenario analyses. First, we implemented a utility decrement to consider the side-effects of bDMARDs since patients treated with these agents may have a lower quality of life than patients receiving csDMARDs only. After mapping SF-36 values from Gerhold et al. to EQ-5D [47,48], we calculated a decrement of 0.0567 for standard care (REM, LDA, M/HDA) and 0.0284 for tapering (TP_REM, TP_LDA). Second, because persistence rates in RA treatment vary widely[49], we included scenarios where patients under standard care will withdraw bDMARDs on their own after two, three or five years of stable remission. Third, we restricted eligibility for de-escalation to one time only. Finally, we varied the time horizon of our model to five, 10, 20 and 40 years.

# 3 Results

## 3.1 Base case

After 120 cycles (i.e., 30 years), our base case model yielded negative incremental costs and negative incremental QALYs for all three de-escalation approaches compared to standard care. Incremental costs per patient were -€78,845 for tapering, -€121,691 for withdrawal and -€107,696 for dose reduction followed by withdrawal, while incremental QALYs were -0.1498, -0.5611 and—0.4354, respectively. The incremental cost-effectiveness ratios (ICERs)–i.e. the costs saved per QALY lost–were €526,254 for tapering, €216,879 for withdrawal, and €247,987 for dose reduction followed by withdrawal.

## 3.2 Sensitivity analyses

Our results remained largely unaffected by changes in parameters related to costs and quality of life (Fig 2). Incremental QALYs were only sensitive to changes in transition probabilities under remission. An increase beyond 14.6% (base case: 7.9%) in the probability of moving from TP_REM to M/HDA shifted our results to being dominant for tapering. Reducing the efficacy of standard care by applying a higher probability of 9.1% (base case: 4.9%) for moving from REM to M/HDA also rendered tapering as the dominant strategy (Fig 2). Pharmaceutical costs for bDMARDs are the most influential cost parameter for all three models. Application of the upper (lower) bound resulted in negative incremental costs of up to €179,002 when bDMARDs are withdrawn (€64,371). We further investigate the impact of setting the upper and lower bounds of bDMARD costs to the most extreme points of our chosen gamma

**Table 2. Model parameters.** Base case model parameters, upper and lower bounds for deterministic sensitivity analysis and distribution parameters for probabilistic sensitivity analysis.

| Model Parameter | Base case | Deterministic | | Probabilistic | | | Source |
|---|---|---|---|---|---|---|---|
| | | Low | High | | mean | SD | |
| Annual costs [€] | | | | | | | |
| Pharmaceutical costs | | | | | | | |
| bDMARDs | 14,834 | 8,147 | 21,520 | Gamma | 14,834 | 6,687 | [43,45] |
| DMARDs | 1,086 | 596 | 1,575 | Gamma | 1,086 | 490 | [43,45] |
| Other (NSAIDs, glucocorticoids) | 45 | 25 | 65 | Gamma | 45 | 20 | [43] |
| Other direct costs (18–64 years) | | | | | | | |
| REM | 1,495 | 821 | 2,168 | Gamma | 1,760 | 793 | [5,45] |
| LDA | 2,316 | 1,272 | 3,360 | Gamma | 2,386 | 1,068 | [5,45] |
| M/HDA | 3,713 | 2,039 | 5,387 | Gamma | 3,797 | 1,712 | [5,45] |
| Other direct costs (>65 years) | | | | | | | |
| REM | 2,024 | 1,112 | 2,936 | Gamma | 1,760 | 793 | [5,45] |
| LDA | 2,421 | 1,330 | 3,512 | Gamma | 2,386 | 1,068 | [5,45] |
| M/HDA | 3,881 | 2,132 | 5,630 | Gamma | 3,797 | 1,712 | [5,45] |
| Transition probabilities | | | | | | | |
| All-cause mortality at cycle one | | | | | | | |
| REM/LDA | 0.0008 | . | . | . | | | |
| M/HDA | 0.0019 | . | . | . | | | |
| Standard therapy | | | | | | | |
| REM → REM | 0.8911 | 0.8457 | 0.9585 | Dirichlet | (111, 8, 6) | | [19–21] |
| REM → LDA | 0.0666 | 0.0039 | 0.0910 | Dirichlet | (111, 8, 6) | | [19–21] |
| REM → M/HDA | 0.0491 | 0 | 0.1048 | Dirichlet | (111, 8, 6) | | [19–21] |
| LDA → REM | 0.3291 | 0.1702 | 0.4879 | Dirichlet | (126, 245, 10) | | [22–25] |
| LDA → LDA | 0.6390 | 0.5079 | 0.7700 | Dirichlet | (126, 245, 10) | | [22–25] |
| LDA → M/HDA | 0.0256 | 0.0080 | 0.0431 | Dirichlet | (126, 245, 10) | | [22–25] |
| M/HDA → REM | 0.1760 | 0.1290 | 0.2219 | Dirichlet | (843, 429, 3499) | | [22,25–31,34–40] |
| M/HDA → LDA | 0.0895 | 0.0593 | 0.1196 | Dirichlet | (843, 429, 3499) | | [22,25–31,34–40] |
| M/HDA → M/HDA | 0.7305 | 0.6609 | 0.8015 | Dirichlet | (843, 429, 3499) | | [22,25–31,34–40] |
| Tapering (TP) | | | | | | | |
| TP_REM → TP_REM | 0.8733 | 0.7562 | 0.9903 | Dirichlet | (55, 7, 5) | | [21] |
| TP_REM → LDA | 0.1111 | 0.0085 | 0.2138 | Dirichlet | (55, 7, 5) | | [21] |
| TP_REM → M/HDA | 0.0791 | 0.0125 | 0.1458 | Dirichlet | (55, 7, 5) | | [21] |
| TP_LDA → TP_REM | 0.2547 | 0.1946 | 0.3148 | Dirichlet | (45, 126, 5) | | [22,24] |
| TP_LDA → TP_LDA | 0.7216 | 0.6598 | 0.7834 | Dirichlet | (45, 126, 5) | | [22,24] |
| TP_LDA → M/HDA | 0.0237 | 0.0027 | 0.0447 | Dirichlet | (45, 126, 5) | | [22,24] |
| Withdrawal (WD) | | | | | | | |
| WD_REM → WD_REM | 0.7394 | 0.6411 | 0.8378 | Dirichlet | (136, 16, 30) | | [19–21,31] |
| WD_REM → LDA | 0.0843 | 0.0320 | 0.1367 | Dirichlet | (136, 16, 30) | | [19–21,31] |
| WD_REM → M/HDA | 0.1616 | 0.1012 | 0.2220 | Dirichlet | (136, 16, 30) | | [19–21,31] |
| WD_LDA → WD_REM | 0.1764 | 0.0871 | 0.2656 | Dirichlet | (59, 243, 30) | | [22,23,32] |
| WD_LDA → WD_LDA | 0.7322 | 0.6897 | 0.7746 | Dirichlet | (59, 243, 30) | | [22,23,32] |
| WD_LDA → M/HDA | 0.0905 | 0.0086 | 0.1725 | Dirichlet | (59, 243, 30) | | [22,23,32] |
| Utilities per cycle | | | | | Alpha | Beta | |
| REM | 0.75 | 0.69 | 0.81 | Beta | 0.7133 | 0.2378 | [44,46] |
| LDA | 0.71 | 0.68 | 0.74 | Beta | 0.8112 | 0.3313 | [44,46] |
| M/HDA | 0.60 | 0.57 | 0.64 | Beta | 0.8984 | 0.5990 | [44,46] |
| Death | 0 | . | . | . | . | . | |

(*Continued*)

**Table 2.** (Continued)

| Model Parameter | Base case | Deterministic | | Probabilistic | | | Source |
|---|---|---|---|---|---|---|---|
| | | Low | High | | mean | SD | |
| Time horizon | 60 | 40 | 80 | | | | |

LDA: low disease activity, M/HDA: medium or high disease activity, REM: clinical remission, SA: single-arm trial, TP: tapering, WD: withdrawal

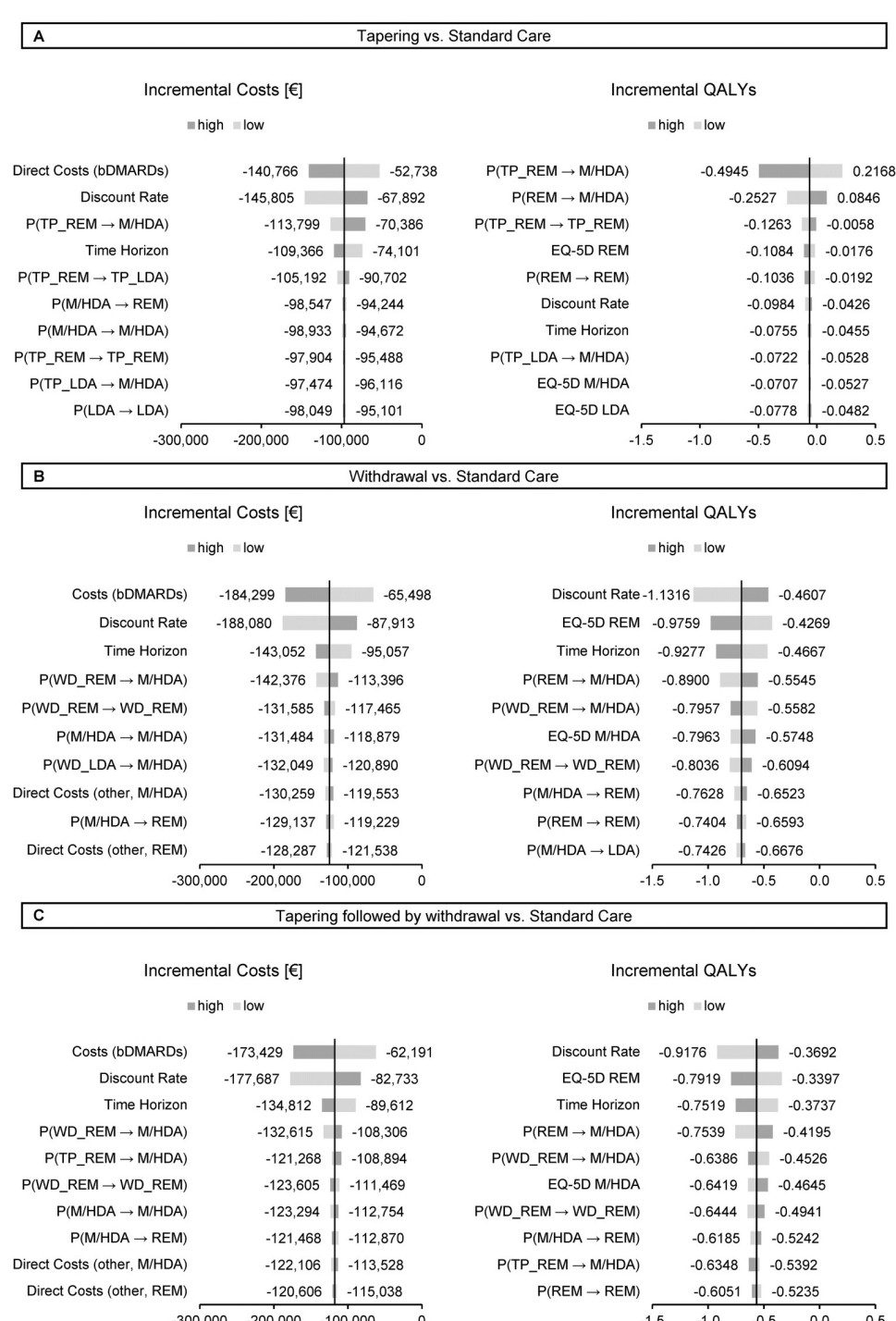

**Fig 2. Results of deterministic sensitivity analyses for incremental costs and quality-adjusted life years (QALYs).**

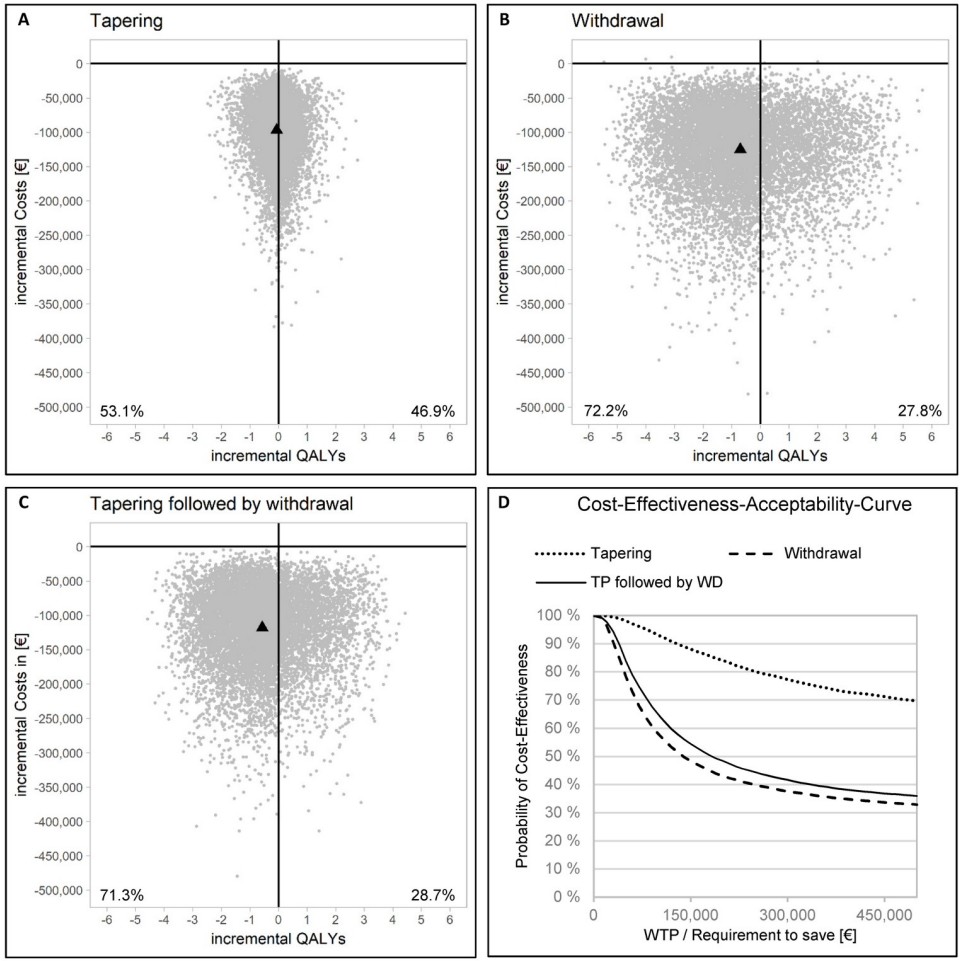

**Fig 3. Results of Monte Carlo simulations depicted in incremental-cost-effectiveness-ratio (ICER) planes and cost-effectiveness-acceptability curves (CEAC).**

distribution (1%/99%-quantiles). Doing so results in negative incremental costs of up to €291,832 for the upper bound and €28,489 for the lower bound.

In our Monte Carlo simulation, tapering, withdrawal, and dose reduction followed by withdrawal were the dominant strategy in 39.8%, 28.2% and 29% of 10,000 iterations, whereas de-escalation was less costly but also less effective in 60.2%, 71.8% and 71% of 10,000 iterations, respectively (Fig 3, panel A-C). The cost-effectiveness-acceptability-curve (CEAC, Fig 3, panel D) shows the probability of a cost-effective outcome (i.e. ICER values smaller than willingness-to-pay (WTP)) for our model results. With model results being less costly and less effective the WTP resembles a requirement to save by the payer. The probability of being cost-effective decreases with increasing requirements to save per QALY lost.

### 3.3 Scenario and structural sensitivity analyses

When we applied a utility decrement for the side effects of bDMARDs, tapering became the dominant strategy (Table 3). For withdrawal and dose reduction followed by withdrawal, the savings generated per QALY lost were substantially increased. When we considered

**Table 3. Results from scenario and structural sensitivity analyses.**

| | Base case | Utility decrement | Self-withdrawal under standard care or dose reduction after... | | | One time de-escalation | Time horizon | | | |
| --- | --- | --- | --- | --- | --- | --- | --- | --- | --- | --- |
| | | | 2 years | 3 years | 5 years | | 5 years | 10 years | 20 years | 40 years |
| Tapering | | | | | | | | | | |
| Δ Costs [€] | -78,845 | -78,844 | -44,502 | -52,506 | -60,886 | -10,260 | -15,880 | -33,395 | -60,511 | -89,101 |
| Δ QALYs | -0.1498 | 0.1588 | -0.0488 | -0.0710 | -0.0952 | -0.0204 | -0.0236 | -0.0560 | -0.1089 | -0.1801 |
| ICER [€] | 526,254 | dominant | 911,350 | 739,387 | 639,706 | 503,255 | 671,627 | 596,880 | 555,532 | 494,695 |
| Withdrawal | | | | | | | | | | |
| Δ Costs [€] | -121,691 | -121,691 | -74,117 | -85,447 | -97,142 | -14,830 | -25,247 | -51,649 | -92,939 | -138,837 |
| Δ QALYs | -0.5611 | -0.0751 | -0.3219 | -0.3788 | -0.4376 | -0.0648 | -0.0766 | -0.1823 | -0.3762 | -0.7398 |
| ICER [€] | 216,879 | 1,620,847 | 230,261 | 225,588 | 222,012 | 229,030 | 329,441 | 283,282 | 247,057 | 187,663 |
| Dose reduction followed by withdrawal | | | | | | | | | | |
| Δ Costs [€] | -107,969 | -107,969 | -60,395 | -71,725 | -83,420 | -13,404 | -21,690 | -45,363 | -82,332 | -123,148 |
| Δ QALYs | -0.4354 | -0.0061 | -0.1962 | -0.2531 | -0.3118 | -0.0515 | -0.0557 | -0.1395 | -0.2920 | -0.5727 |
| ICER [€] | 247,987 | 17,737,935 | 307,882 | 283,437 | 267,516 | 260,234 | 389,613 | 325,098 | 281,981 | 215,039 |

self-withdrawal after two, three or five years of stable remission, savings and the loss of QALYs associated with the three interventions compared to standard care were reduced. The ICER, however, remained similar to the base case. Restricting patients' eligibility for de-escalation to one time only resulted in a decrease in incremental costs saved, as well as a decrease in incremental QALYs lost.

## 4 Discussion

We conducted a cost-utility analysis from the payer's perspective for three different approaches to de-escalating bDMARD treatment in patients with RA: tapering, withdrawal, and tapering followed by withdrawal. The results of our model suggest that all three approaches lead to high cost savings but also a decrease in quality of life. The ICERs appeared to be very high, with €526,254 savings per QALY lost for tapering, €216,879 for withdrawal and €247,987 for tapering followed by withdrawal. Our results remained stable when we varied our input parameters, relaxed assumptions about adherence, or restricted the eligibility of patients for de-escalation to one time only.

To our knowledge, this is the first study to simultaneously (a) consider three different approaches to de-escalating bDMARDs, (b) pool all relevant available evidence on this subject and (c) apply a long-term perspective of 30 year from the payer's perspective. We dealt with the likely heterogeneity of results in published efficacy studies by pooling the available evidence. Our modelling approach can be transferred easily to different health systems by adapting the cost and utility parameters according to the individual system characteristics.

Overall, our results are in line with those of three recent studies[14–16], which found that de-escalating bDMARDs was associated with high cost-savings but also a decrease or only a very slight increase in quality of life compared to continued treatment. Tran-Duy et al.[15], found that savings of €7,133 were associated with a 0.022 loss in QALYs for a time horizon of one year when comparing withdrawal of bDMARDs with continued treatment in a Dutch population. While we found lower savings per year (€4,056) based on our 30-year time horizon, we found a similar loss in quality of life (0.0187). In a French cohort study with a time horizon of 18 months, tapering TNFi-type bDMARDs by means of a spacing strategy was also

associated with saved costs (€8,440) and a loss in QALYs (0.158)[14]. The most similar approach to this in our model–i.e., tapering in the form of a 50% dose reduction followed by withdrawal compared to standard care–was associated with lower cost-savings (€3,225) per year, as well as fewer QALYs lost (0.0189). In a Swedish population, the dose-reduction strategy dominated continued full-dose treatment with etanercept with savings of €6,321 and QALYs gained of 0.010 over a 10 year time horizon[16]. We also identified savings from our tapering strategy after 10 years, though these were substantially higher (€33,395) and accompanied by a small loss of QALYs of 0.0560.

The withdrawal approach in our model showed an ICER of €216,879 after 30 years. This is in contrast to a much higher ICER of €368,269 calculated by Tran-Duy et al. based on the results of the POET trial only, albeit using one-year time horizon and the societal perspective [15]. However, depending on the time horizon, Kobelt et al. found a higher ICER of €262,770 (2 years) and a lower ICER of €81,818 (10 years) when comparing continuous treatment with withdrawal from the societal perspective[16]. When we applied a utility decrement, tapering became the dominant strategy which is in line with results found for a Swedish population [16]. The difference between our ICERs and those in other studies[14–16] is likely driven by the different evidence used to model effectiveness: whereas the other three publications relied on the results of one or only a few RCTs, we considered the pooled results of 22 clinical studies.

To calculate cost-effectiveness, one would usually compare the ICER against WTP threshold for a QALY. Although there is no official threshold of this type in Germany, we can apply the thresholds suggested by the World Health Organization (WHO) [50] or WTP per QALY for Germany calculated by Ahlert et al.[51]. If judged against these, our results can be regarded as highly cost-effective. Judging against the WTP per QALY in Germany of €18,420 [51] we find a probability of being cost-effective of 98% for all three interventions. Applying the threshold of one to three times the GDP suggested by the WHO[50] shows 92%/74% for tapering, 90%/60% for withdrawal and 91%/62% for tapering followed by withdrawal (Fig 3, panel D).

However, the WTP for a positive outcome might not match the willingness to accept (WTA) a negative consequence[52]. For the healthcare sector, O'Brien et al.[52] found a WTA/WTP ratio of 1.9 for the introduction of a new drug, whereas Carthy et al.[53] reported a ratio of 6.9 when assessing the WTA/WTP ratio for a two-week hospital stay. If we consider a hypothetical scenario in which the most extreme WTP value calculated by Ahlert et al.[51] (€18,420) is multiplied by the WTA/WTP ratio of 6.9, our model results still suggest that there is a high probability that de-escalation is cost-effective (tapering: 73%, withdrawal: 58%, dose reduction followed by withdrawal: 60%).

Among the three de-escalation strategies we modelled, tapering achieved the highest savings per QALY lost. This would also meet recommendations by international guidelines to reduce drug dose when patients achieve sustained low disease activity[3,8]. In addition, an immediate withdrawal of bDMARDs would be unlikely to gain acceptance due to risk aversion among patients and rheumatologists. Patients with RA have been found to be reluctant to change their treatment (e.g., to try a new drug, modify the dose of an existing treatment, or take a so-called drug holiday) and have expressed fears of losing disease control or experiencing flares and side effects[54].

Our model has a number of important limitations. First, we did not distinguish between DAS28 calculated using ESR or CRP. Considering the Simplified or Clinical Disease Activity Index (SDAI/CDAI) might increase the clinical accuracy of our estimated transition probabilities. We still chose to include studies that used the DAS28 based on either biomarker because (a) there is a very high correlation of the DAS28 and the SDAI/CDAI scores[55] and

(b) restricting these to SDAI/CDAI only would have substantially reduced the amount of evidence at hand for estimating the parameters of our model.

Second, we had to combine medium and high disease activity into one state (M/HDA) because the underlying studies rarely reported the number of patients within these states. For the same reason, we chose a cut-off of DAS28 < 3.2 to identify LDA in clinical studies, following the classification of Schett et al.[11].

Third, our models may underestimate direct costs (except pharmaceutical costs) because we rely on cost data from registries that, in turn, might be subject to underreporting. This is also true for costs arising from side-effects of concomitant treatments. However, because pharmaceutical costs in our model are four to ten times higher than other direct costs, any underestimation of the latter is very unlikely to jeopardize our results. In the same context, we may overestimate pharmaceutical costs for concomitant medication because we assumed that pharmaceutical costs are independent of disease activity. For example, glucocorticoids or NSAIDs should also be tapered quickly if symptoms have decreased in severity. Again, with costs of less than €0.01 per mg, the impact on our results is negligible.

Fourth, in the base case, we assumed that patients were compliant over the full model horizon of 30 years. While there is evidence that patients and rheumatologists are reluctant to alter their regimen due to the risk of losing disease control, persistence rates in Germany suggest that compliance in patients with RA is imperfect[49]. Regardless, model results were unaffected when we examined possible self-withdrawal of bDMARDs.

Lastly, we modelled dose reduction followed by withdrawal by combining results from studies on tapering with results from studies on immediate withdrawal because only one RCT with 27 patients examined a de-escalation approach consisting of dose reduction followed by withdrawal[21]. In the same context, we include results from RCTs that adjust both csDMARD and/or bDMARD dose in our study pool[21,31]. Additionally, the term "tapering" often refers to a gradual decrease in dose or gradual increase in the spacing of injections, whereas we define it as an immediate 50% reduction in dose. However, modelling individual dosing patterns is beyond the scope of this analysis, and the included RCTs for tapering use the immediate 50% dose reduction only.

## 5 Conclusion

Our findings suggest that de-escalating bDMARDs through tapering (i.e., through an immediate 50% dose reduction), withdrawal, or tapering followed by withdrawal in patients with RA may lead to high cost savings compared with standard care. However, these savings are accompanied by a decrease in quality of life. If decision makers choose to implement de-escalation in daily practice, our results suggest following the tapering approach. In addition, tapering followed by withdrawal might be viable options for patients who want to change their treatment but have not yet done so for fear of losing disease control.

Although our study sheds light on whether de-escalating bDMARDs in patients with RA is a viable option from the payer's perspective, more research using data from clinical practice is necessary to validate our findings.

## Supporting information

**S1 Table. Calculation of pharmaceutical costs.**
(DOCX)

**S2 Table. Detailed overview of study pool for estimation of transition probabilities.**
(DOCX)

**S1 R Code. Model.**

(R)

**S2 R Code. Pooling withdrawal.**

(R)

**S3 R Code. Pooling tapering.**

(R)

**S4 R Code. Pooling standard care.**

(R)

## Author Contributions

**Conceptualization:** Benjamin Birkner, Tom Stargardt.

**Data curation:** Benjamin Birkner.

**Formal analysis:** Benjamin Birkner.

**Methodology:** Benjamin Birkner, Tom Stargardt.

**Project administration:** Tom Stargardt.

**Supervision:** Tom Stargardt.

**Validation:** Benjamin Birkner.

**Visualization:** Benjamin Birkner.

**Writing – original draft:** Benjamin Birkner.

**Writing – review & editing:** Benjamin Birkner, Jürgen Rech, Tom Stargardt.

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
