## [Decision Letter · Decision Letter 0]

14 Aug 2019

PONE-D-19-15210

Cost-utility analysis of bDMARD de-escalation in patients with rheumatoid arthritis

PLOS ONE

Dear Mr. Birkner,

Thank you for submitting your manuscript to PLOS ONE. After careful consideration, we feel that it has merit but does not fully meet PLOS ONE’s publication criteria as it currently stands. Therefore, we invite you to submit a revised version of the manuscript that addresses the points raised during the review process.

We would appreciate receiving your revised manuscript by Sep 28 2019 11:59PM. To enhance the reproducibility of your results, we recommend that if applicable you deposit your laboratory protocols in protocols.io, where a protocol can be assigned its own identifier (DOI) such that it can be cited independently in the future. For instructions see: http://journals.plos.org/plosone/s/submission-guidelines#loc-laboratory-protocols

We look forward to receiving your revised manuscript.

Kind regards,

Mahmoud Abu-Shakra, MD

Academic Editor

PLOS ONE

Additional Editor Comments (if provided):

please address all of the comments of the Reviewers. All statistical analyses need to be revised as it has been suggested by Reviewer 1

Reviewers' comments:

Reviewer's Responses to Questions

**Comments to the Author**

1. Is the manuscript technically sound, and do the data support the conclusions?

Reviewer #1: Partly

Reviewer #2: Yes

2. Has the statistical analysis been performed appropriately and rigorously? 

Reviewer #1: No

Reviewer #2: Yes

3. Have the authors made all data underlying the findings in their manuscript fully available?

Reviewer #1: Yes

Reviewer #2: Yes

4. Is the manuscript presented in an intelligible fashion and written in standard English?

Reviewer #1: Yes

Reviewer #2: Yes

5. Review Comments to the Author

Reviewer #1: Please see comments in the the attached file which discuss issues related to techniques and statistical analysis.

Reviewer #2: This is a new, important and interesting paper. The scientific methodology is valid.

I suggest adding 2 tables for practicing rheumatologist who read the paper: one table that describes all attributes of the studies that were performed on bDMARD de-escalation (setting, number of patients, de-escalation method, result, etc.) and another table on the advantages and disadvantages of bDMARD de-escalation

6. PLOS authors have the option to publish the peer review history of their article (what does this mean?). If published, this will include your full peer review and any attached files.

Reviewer #1: No

Reviewer #2: Yes: Arnon D. Cohen

---

## [Author Response · Author response to Decision Letter 0]

18 Sep 2019

Dear reviewers

Thank you very much for the very constructive feedback on our manuscript. We gratefully included your suggestions and adjusted our model structure to improve our analysis. Please find our replies to all your raised comments in the document called "response to reviewers" which we uploaded together with our revised manuscript.

Kind regards

Benjamin Birkner, Jürgen Rech and Tom Stargardt

---

## [Decision Letter · Decision Letter 1]

12 Nov 2019

PONE-D-19-15210R1

Cost-utility analysis of de-escalating biological disease modifying anti rheumatic drugs in patients with rheumatoid arthritis

PLOS ONE

Dear Mr. Birkner,

Thank you for submitting your manuscript to PLOS ONE. After careful consideration, we feel that it has merit but does not fully meet PLOS ONE’s publication criteria as it currently stands. Therefore, we invite you to submit a revised version of the manuscript that addresses the points raised during the review process.

The manuscript has significantly been improved 

please address the Reviewer comments 

We would appreciate receiving your revised manuscript by Dec 27 2019 11:59PM. To enhance the reproducibility of your results, we recommend that if applicable you deposit your laboratory protocols in protocols.io, where a protocol can be assigned its own identifier (DOI) such that it can be cited independently in the future. For instructions see: http://journals.plos.org/plosone/s/submission-guidelines#loc-laboratory-protocols

We look forward to receiving your revised manuscript.

Kind regards,

Mahmoud Abu-Shakra, MD

Academic Editor

PLOS ONE

Reviewers' comments:

Reviewer's Responses to Questions

**Comments to the Author**

1. If the authors have adequately addressed your comments raised in a previous round of review and you feel that this manuscript is now acceptable for publication, you may indicate that here to bypass the “Comments to the Author” section, enter your conflict of interest statement in the “Confidential to Editor” section, and submit your "Accept" recommendation.

Reviewer #3: (No Response)

2. Is the manuscript technically sound, and do the data support the conclusions?

Reviewer #3: Yes

3. Has the statistical analysis been performed appropriately and rigorously? 

Reviewer #3: Yes

4. Have the authors made all data underlying the findings in their manuscript fully available?

Reviewer #3: No

5. Is the manuscript presented in an intelligible fashion and written in standard English?

Reviewer #3: Yes

6. Review Comments to the Author

Reviewer #3: Here is a list of specific comments. Note: line and page numbering in reviews and comments is based on those in Editorial Manager-generated PDF (cleaned version).

1. Page 4, line 46: Should “our starting cohort” be ‘our hypothetical starting cohort’ because all results were based on simulations?

2. Page 6, lines 81–82: How did the “random effects pooling” fit into the three-step approach mentioned in the next paragraph? If it was part of the three-step approach, I suggest to include the random effects pooling in the three-step approach. If not, I suggest to make the role of random effects pooling more clear and what its relationship with the three-step approach.

3. page 8, Table 1: If I understood the Transition Probabilities section correctly, Table 1 depicted the transition probabilities summarized by the literature review. However, the transition probabilities that were used in the Markov models were estimated from the random effects pooling based on transition probabilities in Table 1. If this is true, I suggest to include the transition probabilities used for the Markov models as Table 1 and the current Table 1 as a supplemental table. Please confirm and clarify in the manuscript.

4. Page 12, Table 2: I think “baseline model” was later referred as “base case” or “base case model”. I suggest to use a consistent name throughout the manuscript.

5. Page 16, line 224: The sensitivity analyses could be more informative if the deterministic low-high bounds were more extreme. Taking bDMARDs of pharmaceutical costs under annual costs, the low (8147) and high (21520) bounds were around 15% and 85% of the associated gamma distribution. I thought more extreme values such as 1% and 99% could be useful to provide the lower and upper limits for the results.

7. PLOS authors have the option to publish the peer review history of their article (what does this mean?). If published, this will include your full peer review and any attached files.

Reviewer #3: No

---

## [Author Response · Author response to Decision Letter 1]

29 Nov 2019

Dear Reviewers, 

please find our replies to all your raised comments in the attached document labelled "Response to Reviewers". 

Best regards

Benjamin Birkner, Jürgen Rech, Tom Stargardt

---

## [Editor Report · Decision Letter 2]

6 Dec 2019

Cost-utility analysis of de-escalating biological disease modifying anti rheumatic drugs in patients with rheumatoid arthritis

PONE-D-19-15210R2

Dear Dr. Birkner,

We are pleased to inform you that your manuscript has been judged scientifically suitable for publication and will be formally accepted for publication once it complies with all outstanding technical requirements.

With kind regards,

Mahmoud Abu-Shakra, MD

Academic Editor

PLOS ONE
---

## [Editor Report · Acceptance letter]

19 Dec 2019

PONE-D-19-15210R2 

Cost-utility analysis of de-escalating biological disease-modifying anti-rheumatic drugs in patients with rheumatoid arthritis 

Dear Dr. Birkner:

I am pleased to inform you that your manuscript has been deemed suitable for publication in PLOS ONE. Congratulations! Your manuscript is now with our production department. 

With kind regards,

on behalf of

Dr. Mahmoud Abu-Shakra 

Academic Editor

PLOS ONE